# Lagrangian simulation and tracking of the mesoscale eddies contaminated by Fukushima-derived radionuclides

**Sergey V. Prants, Maxim V. Budyansky, and Michael Yu. Uleysky**

Laboratory of Nonlinear Dynamical Systems, Pacific Oceanological Institute of the Russian Academy of Sciences, 43 Baltiyskaya st., 690041 Vladivostok, Russia, URL: http://dynalab.poi.dvo.ru

*Correspondence to:* S.V. Prants
(prants@poi.dvo.ru)

**Abstract.** A Lagrangian methodology is developed to simulate, track, document and analyze origin and history of water masses in ocean mesoscale features. It is aimed to distinguish water masses inside the mesoscale eddies originated from the main currents in the Kuroshio – Oyashio confluence zone. By computing trajectories for a large number of synthetic Lagrangian particles advected by the AVISO velocity field after the Fukushima accident, we identify and track the mesoscale eddies which have been sampled in the cruises in 2011 and 2012 and estimate their risk to be contaminated by Fukushima-derived radionuclides. The simulated results are compared with *in situ* measurements showing a good qualitative correspondence.

## 1 Introduction

High tsunami waves after the Tohoku earthquake on March 11, 2011 damaged the cooling system of the Fukushima Nuclear Power Plant (FNPP). Due to lack of electricity, it was not possible to cool nuclear reactors and the fuel storage pools that caused numerous explosions at the FNPP (for details see Povinec et al., 2013). The Fukushima accident was classified at the maximum level of 7, similar to the Chernobyl accident which happened in 1986 in the former Soviet Union. Radionuclides were released from the FNPP through two major pathways, direct discharges of radioactive water and atmospheric deposition onto the North Pacific Ocean. Indirect estimation of that deposition is in the range $6.4$–$35$ PBq (Kumamoto et al., 2014). The total amount of $^{137}$Cs isotope released into the ocean was estimated to be $3.6 \pm 0.7$ PBq by the end of May 2011 (Tsumune et al., 2013).

A few special research vessel (R/V) cruises have been conducted just after the accident and later to measure radioactivity in sea water, zooplankton, fish and in other marine organisms. $^{137}$Cs and $^{134}$Cs isotopes with 30.17 yr and 2.06 yr half-life, respectively, have been detected over a broad area in the western North Pacific in 2011 and 2012 (Honda et al., 2012; Buesseler et al., 2012; Inoue et al., 2012a, b; Tsumune et al., 2012, 2013; Kaeriyama et al., 2013; Oikawa et al., 2013; Aoyama et al., 2013; Kameník et al., 2013; Kumamoto et al., 2014; Kaeriyama et al., 2014; Budyansky et al., 2015). $^{137}$Cs concentration levels off Japan before the accident were estimated at the background level to be 1–3 mBq/kg, while $^{134}$Cs was not detectable. Because of a comparatively short half-life time, any measured concentrations of $^{134}$Cs could only be Fukushima derived.

The studied area is shown in Fig. 1a. It is known as the Kuroshio – Oyashio confluence zone or a subarctic frontal area (Kawai, 1972). The Kuroshio Extension prolongs the Kuroshio Current which turns to the east at about $35°$ N and flows as a strong meandering jet constituting a front separating the warm subtropical and cold subarctic waters. It is a region with one of the most intense air-sea heat exchange and the highest eddy kinetic energy level. The Kuroshio – Oyashio confluence zone is populated with several mesoscale eddies that transfer heat, salt, nutrients, carbon, pollutants and other tracers across the ocean. They originate, besides from the Kuroshio Extension, from the Tsugaru Warm Current, flowing between the Honshu and Hokkaido islands, and from the cold Oyashio Current flowing out of the Arctic along the Kamchatka Peninsula and the Kuril Islands (Fig. 1a). The lifetime of those eddies ranges from a few weeks to a few years.

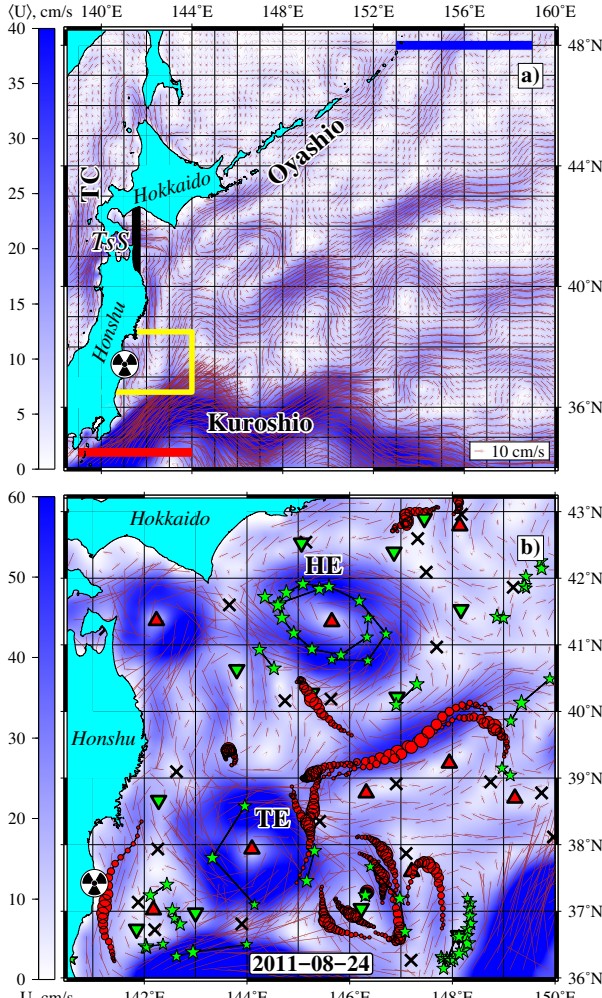

**Figure 1.** a) The AVISO velocity field in the Kuroshio – Oyashio confluence zone averaged from 1993 to 2016. TsS stands for the Tsugaru Strait. Location of the FNPP is shown by the radioactivity sign. Area just around the FNPP is shown by the yellow lines. b) The velocity field on August 24, 2011 with the Tohoku (TE) and Hokkaido (HE) eddies studied in the paper and with tracks of some available drifters (the red circles) and Argo floats (the green stars) to be present in the area at that time. Elliptic and hyperbolic stagnation points with zero mean velocity are indicated by triangles and crosses, respectively, with up- and downward oriented triangles denoting anticyclones and cyclones, respectively.

The standard approach in simulating transport phenomena like propagation of oil after the explosion at the Blue Horizon mobile drilling rig in the Gulf of Mexico in April 2010 and propagation of radioactive isotopes after the accident at the FNPP is to run global or regional numerical models of circulation to simulate propagation of pollutants and try to forecast their trajectories. The outcomes provide "spaghetti-like" plots of individual trajectories which are hard to interpret. Moreover, as majority of real trajectories in a chaotic environment are very sensitive to small and inevitable vari-

ations in initial conditions, they are practically unpredictable even over a comparatively short time.

A specific Lagrangian approach, based on dynamical systems theory, has been developed in the last decades with the aim to find more or less robust material structures in chaotic flows governing mixing and transport of Lagrangian particles and creating transport barriers preventing propagation of a contaminant across them (for reviews see Samelson and Wiggins, 2006; Mancho et al., 2006; Koshel' and Prants, 2006; Haller, 2015). Identification of such structures in the ocean would help to predict for a short and medium time where a contaminant will move even without a precise solution of the Navier – Stokes equations. This approach has been successfully used in simulating propagation of oil in the Gulf of Mexico (Mezić et al., 2010; Huntley et al., 2011; Olascoaga and Haller, 2012) and propagation of Fukushima-derived radionuclides in the Pacific ocean (Prants et al., 2011b; Budyansky et al., 2015; Prants et al., 2014).

The present authors have developed a set of Lagrangian tools for tracking origin, history and fate of water masses advected by analytic, altimetric and numerical velocity fields generated by eddy-resolved regional circulation models (Budyansky et al., 2009; Prants et al., 2011b, a; Prants, 2013; Prants et al., 2013; Prants, 2014; Budyansky et al., 2015). Each elementary volume of water can be attributed to physico-chemical properties (temperature, salinity, density, radioactivity, etc.) which characterize this volume as it moves. In addition, each water parcel can be attributed to other types of diagnostics which are exclusively function of its trajectory. We call them "Lagrangian indicators". They are, for example: distance travelled by a fluid particle for some period of time; absolute, zonal, and meridional displacements of particles from their original positions; the number of their cyclonic and anticyclonic rotations; time of residence of fluid particles inside a given area; exit time out off that area; the number of times particles visited different places in a studied region.

The Lagrangian indicators contain information about the origin, history and fate of the corresponding water masses and allow to the identification of water masses that move coherently, either by propagating together or by rotating together. Even if adjacent waters are indistinguishable, say, by temperature (e.g., the satellite SST images indicate no thermal front), the corresponding water masses could still be distinguishable, for example, by their origin, travelling history and other factors. The Lagrangian indicators are computed by integrating advection equations (1) for a large number of synthetic particles forward and backward in time. When integrating (1) forward in time one computes particle's trajectories to know the fate of the corresponding particles and when integrating (1) backward in time one could know where the particles came from and the history of their travel.

The purpose of this paper is threefold. Firstly, we develop a Lagrangian methodology in order to track and document the origin and history of water masses constituting promin-

ent mesoscale feature. It allows to distinguish water masses inside mesoscale eddies originated from the main currents in the Kuroshio – Oyashio confluence zone. Secondly, we apply that methodology in order to identify and track the mesoscale eddies, advected by the altimetric AVISO velocity field, with a risk to be contaminated by Fukushima-derived radionuclides. Finally, the simulation results are compared qualitatively with *in situ* sampling of those eddies in the R/V cruises. The location and form of the simulated eddies are verified, when possible, by tracks of surface drifters and diving Argo floats available at the sites aoml.noaa.gov/phod/dac and www.argo.net, respectively.

## 2  Data and methodology

All the simulation results are based on integrating equations of motion for a large number of synthetic particles (tracers) advected by the AVISO velocity field

$$\frac{d\lambda}{dt} = u(\lambda, \varphi, t), \qquad \frac{d\varphi}{dt} = v(\lambda, \varphi, t), \qquad (1)$$

where $u$ and $v$ are angular zonal and meridional velocities, $\varphi$ and $\lambda$ are latitude and longitude, respectively. The altimetry-based velocities were obtained from the AVISO database (aviso.altimetry.fr) archived daily on a $1/4° \times 1/4°$ grid. The velocity field has been interpolated using a bicubical spatial interpolation and third order Lagrangian polynomials in time. In integrating Eqs. 1 we used a fourth-order Runge – Kutta scheme with an integration step of 0.001 day.

The velocity field is from altimetry data, which provide the geostrophical component of the real near-surface velocities valid at the mesoscale. In order to display the enormous amount of information, we plot maps of specific Lagrangian indicators versus particle's initial positions. The region under study is seeded with a large number of Lagrangian particles whose trajectories are computed for a given period of time. The results obtained are processed to get a data file with the field of a specific Lagrangian indicator in this area. Finally, its values are coded by color and represented as a map in geographic coordinates.

It is informative also to identify "instantaneous" stagnation elliptic and hyperbolic points on the Lagrangian maps. We mark them by triangles and crosses, respectively. They are points with zero velocity which are computed daily with the AVISO velocity field. The elliptic points are called stable and the hyperbolic ones are unstable. Their local stability properties are characterized by a standard method calculating eigenvalues of the Jacobian matrix of the velocity field. The elliptic points, situated mainly in the centers of eddies, are those points around which the motion is stable and circular. Up(down)ward orientation of one of the triangle's top on the maps means anticyclonic (cyclonic) rotations of water around them. The hyperbolic points, situated mainly between and around eddies, have stable manifolds along which water parcels converge to such a point and unstable manifolds

along which they diverge. The stagnation points are moving Eulerian features and may undergo bifurcations in the course of time. In spite of nonstationarity of the velocity field, some of them may exist for weeks and much more. The hyperbolic points and their attracting and repelling manifolds have been recently identified with the help of drifter's tracks in the Gulf of La Spezia in the northwestern Mediterranean Sea (Haza et al., 2010), in the Gulf of Lion (Nencioli et al., 2011), in the Gulf of Mexico (Olascoaga et al., 2013) and in the northwestern Pacific (Prants et al., 2016).

The altimetry-based Lagrangian maps allow accurately identify and track mesoscale eddies and document their transformation due to interactions with currents and other eddies. Inspecting daily-computed Lagrangian maps for a long period of time (up to two years in this paper) and computing stagnation elliptic points daily, one can track the origin and fate of water masses within a given eddy if it is sufficiently large and long lived (more than a week). For this purpose Lagrangian diagnostics are more appropriate than commonly used Eulerian techniques, because Lagrangian maps are imprints of the history of water masses involved in the vortex motion, whereas vorticity, Okubo – Weiss parameter and similar indicators are only "instantaneous" snapshots (see Olascoaga et al., 2013; Prants et al., 2016, for comparison).

Being motivated by the problem of identification of Fukushima-contaminated waters in the core and at the periphery of persistent mesoscale eddies in the area, we develop in this paper a specific Lagrangian technique oriented to distinguish water masses of a different origin inside the eddies with a risk to be contaminated. With this aim we specify, besides Fukushima-derived waters, water masses originated from the main currents in the Kuroshio – Oyashio confluence zone. The integration has been performed backward in time. We removed from consideration all the particles entered into any AVISO grid cell with two or more corners touching the land in order to avoid artifacts due to the inaccuracy of the altimetry-based velocity field near the coast.

In what follows, we define the "yellow" waters on the maps as those which have a large risk to be contaminated because they came after the accident from the area just around the FNPP enclosed by the yellow straight lines in Fig. 1a and for the period from the day of the accident, March 11, 2011 to May 18, 2011 when direct releases of radioactive isotopes to the ocean and atmosphere stopped. The "red" waters are salty and warm Kuroshio waters. To be more exact, they came from the red zonal line ($34.5°$ N, $139°$ E – $144°$ E) in Fig. 1a crossing the Kuroshio main jet. The "black" waters came from the warm Tsushima Current flowing via the Tsugaru Strait out off the Japan Sea and crossed that strait (the black line with $40°$ N – $43°$ N, $141.55°$ E). The "blue" waters are fresher and colder waters originated from the Oyashio Current and crossed the blue zonal line ($48°$ N, $153°$ E – $159°$ E) shown in Fig. 1a. The "white" waters on the Lagrangian maps have not been specified to be originated from one of the

segments mentioned above. They could reach their places on the maps from anywhere besides those segments.

We are interested in advective transport for a comparatively long period of time, up to two years. It is hardly possible to simulate adequately motion of a specified passive particle in a chaotic flow, but it is possible to reproduce transport of statistically significant number of particles. Our results are based not on simulation of individual trajectories but on statistics for 490,000 Lagrangian particles. We cannot, of course, guarantee that we compute "true" trajectories for individual particles. The description of general pattern of transport for half a million particles is much more robust. However, we do not try to simulate quantitatively concentration of radionuclides or estimate the content of water masses of different origin inside the studied eddies.

## 3 Results

A few mesoscale eddies were present in the studied area to the day of the accident. The cyclonic eddies with the centers, marked by the downward-oriented triangles on the Lagrangian maps, prevailed in the area to the north of the Subarctic Front, the boundary between the subarctic ("blue") and subtropical ("red") waters in Fig. 2. The anticyclonic eddies with the centers, marked by the upward-oriented triangles, prevailed to the south of the front.

The large anticyclonic Tohoku eddy (TE) with the center at around $39°$ N, $144°$ E in March 2011 has been sampled after the accident in the two R/V cruises in June (Buesseler et al., 2012) and July 2011 (Kaeriyama et al., 2013) showing large concentrations of $^{137}$Cs and $^{134}$Cs. The anticyclonic Hokkaido eddy (HE), genetically connected with the TE, was born in the middle of May 2011 with the center at around $40°$ E. After that it captured some contaminated water from the TE. It has been sampled in the end of July 2011 (Kaeriyama et al., 2013).

The anticyclonic Tsugaru eddy (TsE) was genetically connected with the HE. It was born in the beginning of February 2012 with the center at around $41.9°$ N, $148°$ E and captured some contaminated water from the HE. The TsE has been sampled in the R/V "*Professor Gagarinskiy*" cruise on July 5, 2012 to have concentrations of $^{137}$Cs and $^{134}$Cs over the background level at the surface and in intermediate depths (Budyansky et al., 2015). All these eddies will be studied in this section from the Lagrangian point of view in order to simulate and track by which transport pathways they could have gained water masses from the Fukushima area or from other origin and to compare qualitatively the simulation results with *in situ* measurements.

### 3.1 The Tohoku eddy

We tracked with daily-computed Lagrangian maps the birth, metamorphoses and decay of the mesoscale anticyclonic TE.

It was born in the middle of May 2010 with the elliptic point at around $38°$ N, $144°$ E at that time as the result of interaction of a warm anticyclonic Kuroshio ring with a cyclone with mixed Kuroshio and Oyashio core waters. It has interacted with other eddies almost for a year with multiple splitting and merging in the area to the east off the Honshu Island. Just after the accident, it begun to gain "yellow" water from the area around the FNPP with a high risk of contamination. That eddy is clearly seen in earlier simulation just after the accident in Fig. 3b by Prants et al. (2011b) and on the Lagrangian map in Fig. 2a as a red patch labeled as TE with the center at $39°$ N, $144°$ E on March 26, 2011.

The maps in Fig. 2 and in the subsequent figures have been computed as it was explained in Sec. 2. The red color in the core of the TE means that its core water was of subtropical origin. More precisely, the red tracers were advected for two years from the red line segment in Fig. 1a to the current place on the map. In March 2011 "yellow" water, coming from the area around the FNPP with a comparatively high risk to be contaminated, wrapped round the TE. A thin streamer of Tsugaru "black" water, coming from the black line segment in Fig. 1a, wrapped a periphery of the TE to the end of March. "Yellow" waters propagated gradually to the east and south due to a system of currents wrapping around the eddies present in the area. The straight zonal boundary along $36.5°$ N and meridional boundary along $144°$ E, separating water masses of different origin in Fig. 2a on March 26, 2011, are just fragments of the boundary in Fig. 1a restricting the area around the FNPP. These boundaries separate the "yellow" tracers which were present within the area from those ones which have not yet managed to penetrate inside the area for 15 days after the accident.

In April and May 2011 the TE had a sandwich-like structure with the red subtropical core belted with a narrow streamer of Fukushima "yellow" waters which, in turn, was encircled by a red streamer of Kuroshio subtropical water (Fig. 2b). A new eddy configuration appeared to the end of May in Fig. 2b with the TE interacting with a "blue" cyclone with the center at $39.9°$ N, $144.7°$ E and a newborn "yellow" anticyclone which we call the Hokkaido eddy (HE) with the center at $40.4°$ N, $145.5°$ E. The core of that cyclone consisted of a "blue" subarctic Oyashio water with low risk to be contaminated, but the HE core water came from the area around the FNPP with a high risk to be contaminated.

In the course of time the TE moved gradually to the south. Its periphery has been sampled in the beginning of June by Buesseler et al. (2012), and the whole eddy has been crossed in the end of July 2011 by Kaeriyama et al. (2013). Fukushima-derived cesium isotopes have been measured on June 10 and 11 during the R/V "*Ka'imikai-o-Kanaloa*" cruise (Buesseler et al., 2012) along the $144°$ E meridional transect where the cesium concentrations have been found to be in the range from the background level, $C_{137} = 1.4 - 3.6$ mBq/kg (stations 13 and 14), to a high level up to $C_{137} = 173.6 \pm 9.9$ mBq/kg (station 10). The ratio $^{134}$Cs/$^{137}$Cs was close to 1.

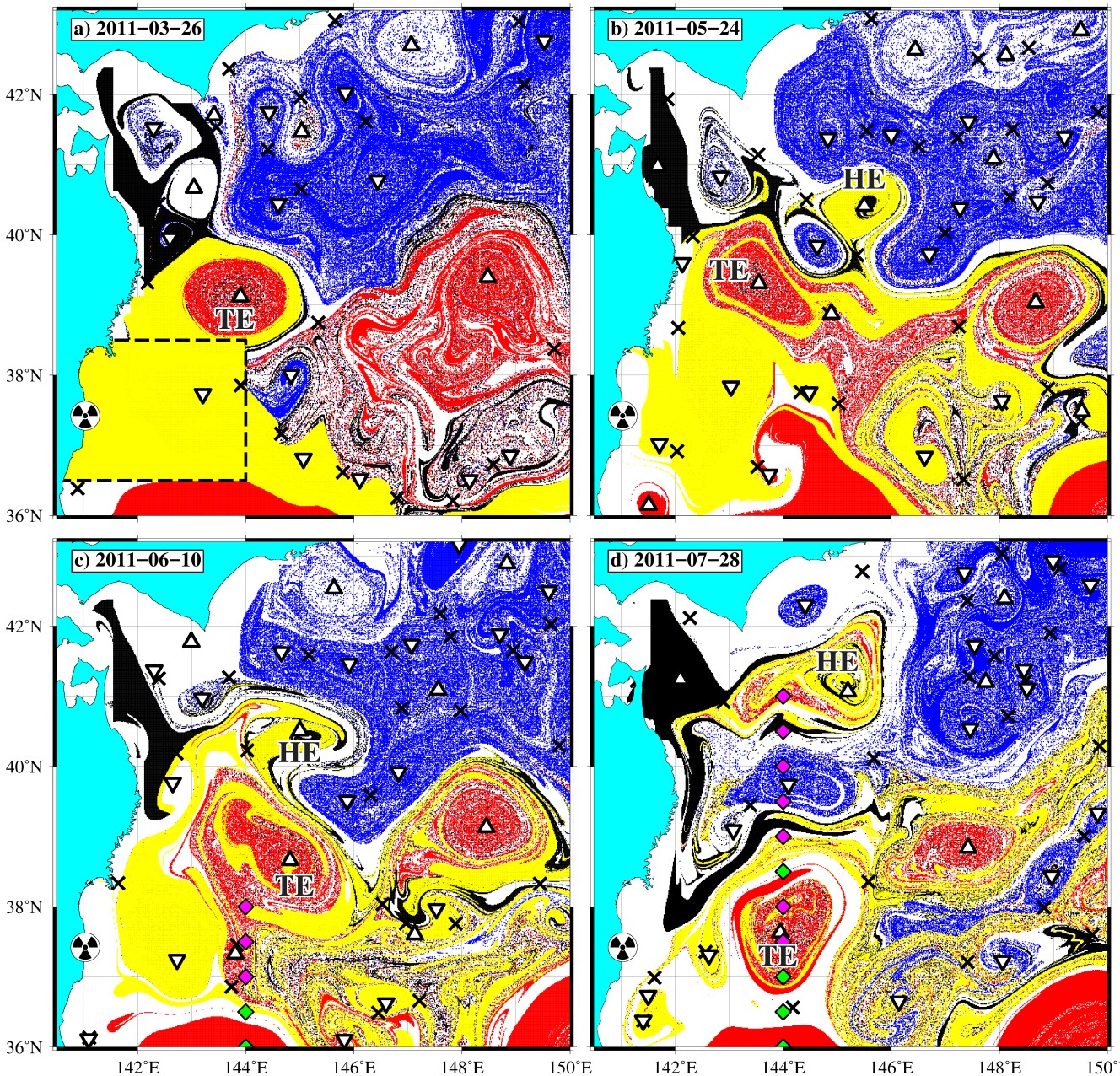

**Figure 2.** The Lagrangian maps show evolution of the Tohoku eddy (TE) after the accident to the days of its sampling and the origin of waters in its core and at the periphery. The red, black and blue colors specify the tracers which came for two years in the past to their places on the maps from the Kuroshio, Oyashio and Tsushima currents, respectively, more exactly, from the corresponding line segments shown in Fig. 1a. The yellow color marks the Lagrangian particles coming from the area around the FNPP in Fig. 1a (shown in Fig. 2a by the dashed line), after the day of the accident on March 11, 2011. The TE has been sampled on June 10 and 11, 2011 by Buesseler et al. (2012) along the transect $35.5°\text{N} - 38°\text{N}$, $144°\text{E}$ shown in panel c) and in the end of July 2011 by Kaeriyama et al. (2013) along the transect $35°\text{N} - 41°\text{N}$, $144°\text{E}$ shown in panel d). The locations of stations with collected by Buesseler et al. (2012) and (Kaeriyama et al., 2013) surface seawater samples with measured radiocesium concentrations at the background level are indicated by the green diamonds. Stations, where the concentrations have been measured to be much higher, are marked by the magenta diamonds.

For ease of comparison, we mark in Fig. 2c by the green diamonds the locations of stations 13 and 14 with collected surface seawater samples by Buesseler et al. (2012) in which the cesium concentrations have been measured to be at the background level ($\lesssim 3.6$ mBq/kg). The stations 10, 11 and 12, where the concentrations have been found to be much

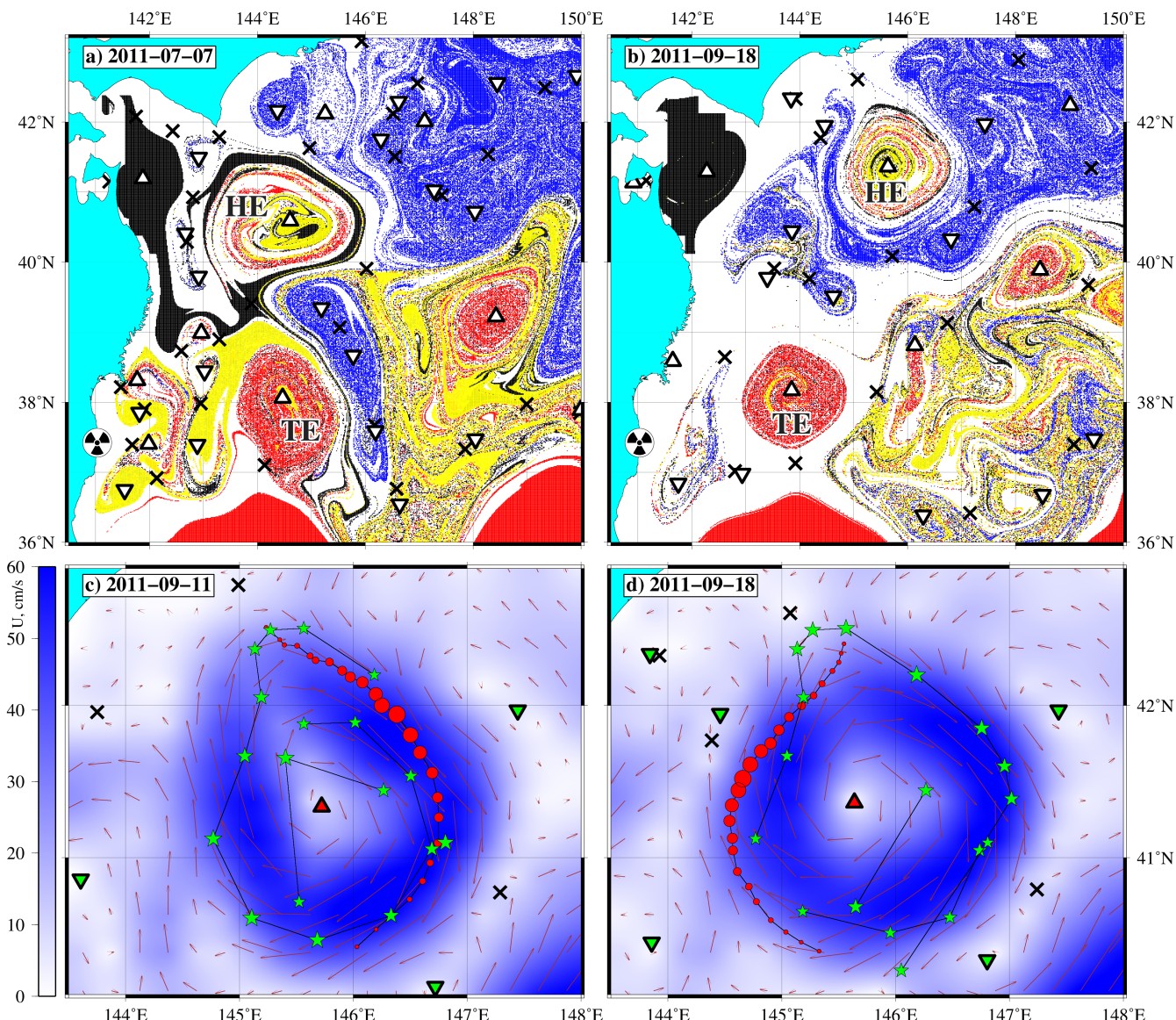

**Figure 3.** a) and b) The Lagrangian maps show evolution of the Hokkaido eddy (HE) after the FNPP accident to the days of its sampling and the origin of waters in its core and at the periphery. c) and d) A fragment of the track of the drifter no. 39123 is indicated by the full circles for two days before the day indicated with the size of circles increasing in time. Tracks of three Argo floats are shown by the stars. The largest star corresponds to the day indicated and the other ones show float positions each 5 days before and after that date.

larger, are indicated by the magenta diamonds. Our simulation in Fig. 2c shows that stations 13 and 14 on the days of sampling have been located in "red" and "white" waters with a low risk to contain Fukushima-derived radionuclides.

Transport and mixing at and around stations 10, 11 and 12 with high measured values of the cesium concentrations by Buesseler et al. (2012) have been governed mainly by the interaction of the TE with the "yellow" mesoscale cyclone with the center at 37.2° N, 142.8° E. This cyclone formed in the area in April and captured "yellow" waters with a high risk of contamination. Unfortunately, it has not been sampled in the R/V "*Ka'imikai-o-Kanaloa*" cruise. The surface seawa-

ter samples at stations 10, 11 and 12, have been collected on the days of sampling at the eastern periphery of that cyclone and at the southern periphery of the TE with the "yellow" streamer there. Station 10 with the highest measured level of the $^{137}$Cs concentration, $C_{137} = 173.6 \pm 9.9$ mBq/kg, was located at 38° N, 144° E inside the wide streamer of "yellow" water around the TE. Stations 11 and 12 with $C_{137} = 103.7 \pm 5.9$ mBq/kg and $C_{137} = 93.6 \pm 4.9$ mBq/kg, respectively, have been located within the narrow streamers with "yellow" simulated water in Fig. 2c intermitted with narrow streamers of "red" water. So, we estimate the risk to find Fukushima-derived radionuclides there (the magenta

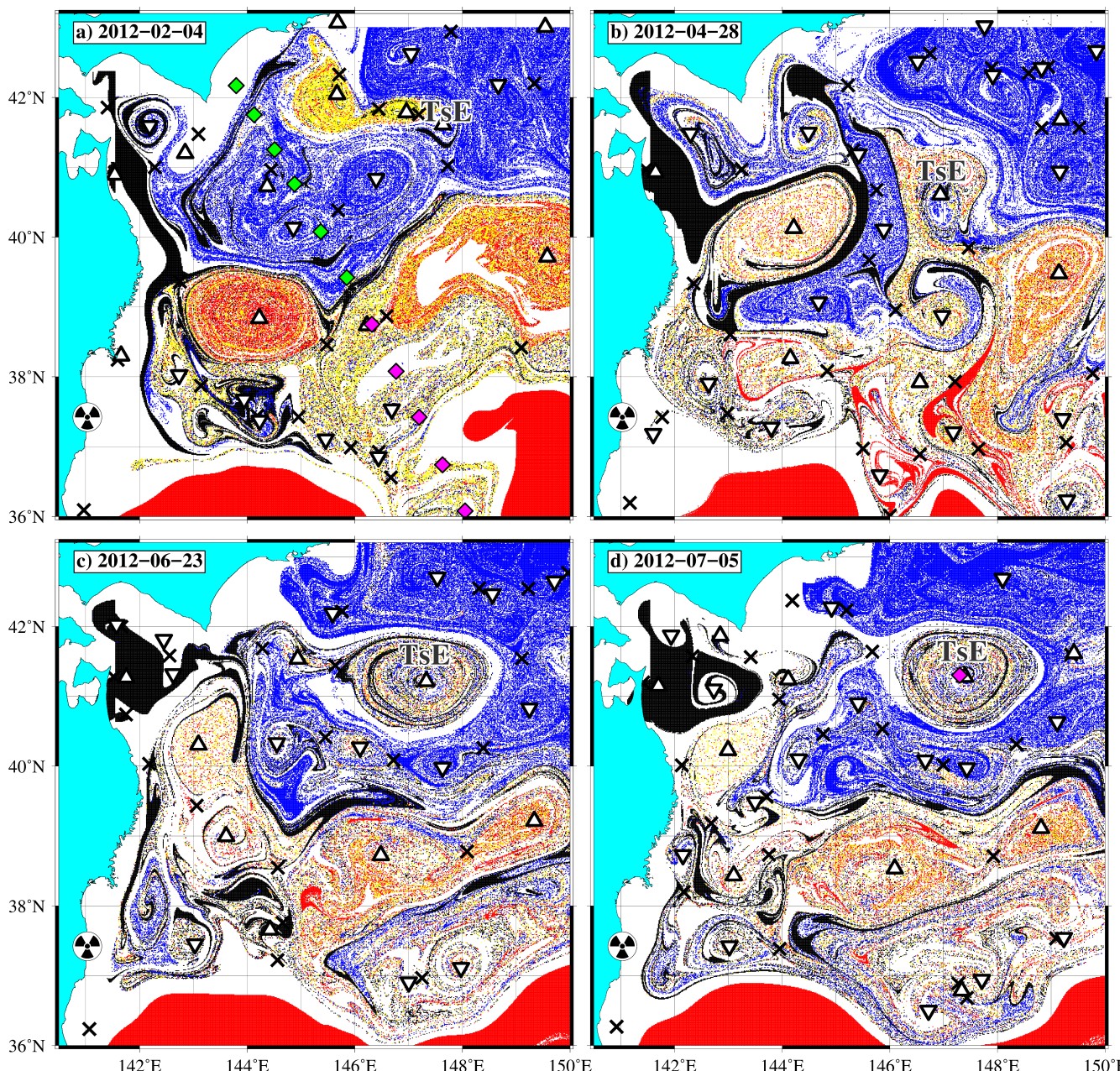

**Figure 4.** The Lagrangian maps in the study area in the first half of 2012. a) The locations of stations in the beginning of February with collected by Kumamoto et al. (2014) surface seawater samples with measured radiocesium concentrations at the background level (the green diamonds) and with higher concentration levels (the magenta diamonds). b) – d) The Lagrangian maps show evolution of the Tsugaru eddy (TsE) which was born on February 4, 2012 (panel a) after splitting of the HE and sampled by Budyansky et al. (2015) at station 84 on July 5, 2012 to have increased radiocesium concentrations (the magenta diamond in panel d).

diamonds) to be much higher than at stations 13 and 14 (the green diamonds) and it is confirmed by a qualitative comparison with measured data.

A specific configuration of mesoscale eddies occurred in the area to the northeast of the FNPP to the end of July 2011, the days of sampling by Kaeriyama et al. (2013) along the $144°$ E meridian from $35°$ N to $41°$ N in the R/V "*Kaiun*

*maru*" cruise. That transect is shown in Fig. 2d. It crosses the TE and the cyclone with "blue" Oyashio water, which is genetically linked to the "blue" cyclone at $39.9°$ N, $144.7°$ E in Fig. 2b. The transect also crosses partly the periphery of the anticyclonic HE. The measured [137]Cs concentrations in surface seawater samples at the stations C43–C55 have been found to be in the range from the background level,

$1.9 \pm 0.4$ mBq/kg, (station C52) to a much higher level of $153 \pm 6.8$ mBq/kg (station C47). The colored tracking maps in Fig. 5 by Prants et al. (2014) show where the simulated tracers of that transect were walking from March 11 to April 10, 2011 being advected by the AVISO velocity field.

The risk of radioactive contamination of the markers placed at $36° \mathrm{N} - 36.5° \mathrm{N}$ was estimated by Prants et al. (2014) to be small, because they have been advected mainly by the Kuroshio Current from the southwest to the east (the corresponding concentrations have been measured by Kaeriyama et al. (2013) to be 2–5 mBq/kg). The present simulation in Fig. 2d also shows that stations C51, 52 and 53 (the green diamonds) with the measured cesium concentrations at the background level on the days of sampling by Kaeriyama et al. (2013) have been located in the "red" waters (stations C51 and C53) advected by the main Kuroshio jet from the southwest and in the "white" waters (station C52) between the TE and the jet. Therefore, we estimate a risk to find Fukushima-derived radionuclides there to be comparatively low.

The transect $36.5° \mathrm{N} - 38° \mathrm{N}$ in Fig. 2d (the red one in Fig. 5 by Prants et al. (2014)) crossed the TE. The $^{137}$Cs concentrations at the stations C49 and C50 of that transect have been measured to be $36 \pm 3.3$ and $50 \pm 3.6$ mBq/kg (Kaeriyama et al., 2013). Comparing those results with simulated ones, we note the presence of "yellow" water in the TE core at the locations of those stations. Surface samples at station C48 ($38.5° \mathrm{N}$) have been measured to contain the $^{137}$Cs concentration to be at the background level $2.7 \pm 0.6$ mBq/kg (Kaeriyama et al., 2013). The corresponding green diamond is located in our simulation in the area with "red" and "white" waters.

Inspecting the Lagrangian maps on the days between June 6 and July 28 (not shown), we have found that the "yellow" cyclone with the center at $37.2° \mathrm{N}$, $142.8° \mathrm{E}$ in Fig. 2c collapsed in the end of June. Its "yellow" core water with a high risk to be contaminated has been wrapped around the neighbor anticyclone TE in the form of a wide yellow streamer visible in Fig. 2d. The highest concentration $C_{137} = 153 \pm 6.8$ mBq/kg has been measured by Kaeriyama et al. (2013) at station C47 ($39° \mathrm{N}$) situated in the area of that streamer. Stations C46 ($39.5° \mathrm{N}$) with $C_{137} = 83 \pm 5.0$ mBq/kg is situated in the close proximity to a yellow streamer sandwiched between "white" and "black" waters. Comparatively high concentration $C_{137} = 65 \pm 4.3$ mBq/kg has been measured by Kaeriyama et al. (2013) at station C45 ($40° \mathrm{N}$) that was on the days of sampling in the core of the "blue" cyclone with the center at $39.7° \mathrm{N}$, $144.2° \mathrm{E}$ (Fig. 2d). Our simulation shows that it has been formed mainly by Oyashio "blue" waters (with a low risk to be contaminated by Fukushima-derived radionuclides) and partly by "white" waters.

When comparing simulation results in Fig. 2d with the measurements by Kaeriyama et al. (2013), we have found that the simulation is consistent with samplings at stations C48, 51, 52 and 53 in the sense that the cesium concentrations have been measured to be at the background level in those places on the maps where there is no signs of "yellow" water with a high risk to contain Fukushima-derived radionuclides. Our simulation is consistent at least quantitatively also with samplings at stations C47, 49 and 50 with high measured levels of the cesium concentrations because the "yellow" water presents there in our simulation.

However, there is an inconsistency of simulation with samplings at stations C45 and C46 where there are practically now yellow tracers but only blue and white ones. The reasons of this inconsistency might be different. In this paper we track only those tracers which were originated from the blue, red and black segments and the yellow rectangular around the FNPP shown in Fig. 1a. So we did not specify the origin of white waters. They could reach their places on the maps from anywhere besides those segments and the area around the FNPP. They could in principle contain Fukushima-derived radionuclides deposited at the sea surface from the atmosphere after the accident and then being advected by eddies and currents in the area. Moreover, they could be those tracers which have been located inside AVISO grid cells near the coast around the FNPP just after the accident and then have been advected outside. We removed from consideration all the tracers entered into any AVISO grid cell with two or more corners touching the land because of inaccuracy of the altimetry-based velocity field there and in order to avoid artifacts.

Thus, the white streamers inside the core and at the periphery of the blue cyclone with the center at $39.7° \mathrm{N}$, $144.2° \mathrm{E}$ nearby stations C45 and C46 with high measured concentrations of cesium by Kaeriyama et al. (2013), could, in principle, contain contaminated water. However, it has not been proved in our simulation by the mentioned reasons.

## 3.2  The Hokkaido eddy

Now we consider the anticyclonic HE. It was born in the middle of May (see the yellow patch in Fig. 2b with the center at $40.3° \mathrm{N}$, $145.5° \mathrm{E}$) being genetically linked to the TE. During May, the TE gradually lost a Fukushima "yellow" water from its periphery to form the core of the HE. Fig. 3a shows the HE with a yellow core surrounded by modified subtropical "red" water which, in turn, is surrounded by Tsugaru "black" water.

The sampling of that eddy and its periphery by Kaeriyama et al. (2013) along the $144° \mathrm{E}$ meridian in the end of July showed comparatively high concentrations, $C_{137} = 60 \pm 4.0$ and $71 \pm 4.6$ mBq/kg at stations C44 ($40.5° \mathrm{N}$) and C43 ($41° \mathrm{N}$), respectively. Station C43 was located inside the anticyclone HE filled mainly by "yellow" waters, and we estimate the risk to found Fukushima-derived radionuclides there to be large. Station C44 was located at the southern periphery of the anticyclone HE at the boundary between "white" and "blue" waters but in close proximity to a "yellow" streamer.

The location of the HE on August 24, 2011 is shown in the AVISO velocity field in Fig. 1b. To verify the simulated locations of the HE and its form, we plot in Figs. 3c and d fragments of the tracks of a drifter and three Argo floats captured by that eddy in September 2011. A fragment of the track of the drifter no. 39123 is shown by the red circles with the size increasing in time for two days before the dates indicated in Figs. 3c and d and decreasing for two days after those dates, i.e. the largest circle corresponds to the drifter position at the indicated date. It was launched after the accident on July 18, 2011 at the point 45.588° N, 151.583° E in the Oyashio Current, advected by the current to the south and eventually captured by the HE moving around clockwise. Fragments of the clockwise tracks of the three Argo floats are shown in Figs. 3c and d by stars for seven days before and seven days after the indicated dates. The float no. 5902092 was released long before the accident on September 9, 2008 at the point 32.699° N, 145.668° E to the south off the Kuroshio Extension jet and was able to cross the jet and to get far north. The float no. 2901019 was released before the accident on April 19, 2010 at the point 41.723° N, 146.606° E. The float no. 2901048 was released just after the accident on April 10, 2011 at the point 37.469° N, 141.403° E nearby the FNPP.

Our simulation shows that the HE contained after its formation in the middle of May 2011 a large amount of a "yellow" water probably contaminated by the Fukushima-derived radionuclides. This conclusion is supported by an increased concentration of radiocesium measured in its core at station C43 by Kaeriyama et al. (2013) in the end of July 2011. The HE persisted in the area around 42° N, 148° E up to the end of January of the next year. It splitted eventually on January 31, 2012 into two anticyclones.

## 3.3 The Tsugaru eddy

The anticyclonic TsE was born on February 4, 2012 after decay of the HE (the yellow patch with the elliptic point at 42° N, 145.6° E in Fig. 4a). The elliptic point at the center of the TsE appeared at 41.8° N, 146.9° E. Just after its birth, the HE begun to transport its "yellow" water around the TsE with the core consisted of an Oyashio "blue" water (Fig. 4b). The strong Subarctic Front is visible in Fig. 4 as a contrast boundary between Oyashio "blue" water and Fukushima-derived "yellow" water with the Tsugaru "black" water in between.

Seawater samples for radiocesium measurements in the frontal area have been collected during the R/V "Mirai" cruise from January 31 to February 5, 2012 along one of observation lines of the World Ocean Circulation Experiment (WOCE) in the western Pacific, specifically the WOCE-P10/P10N line (Kumamoto et al., 2014). We impose on the simulated Lagrangian map in Fig. 4a locations of stations to the north of the Kuroshio Extension (>36° N) with measured levels of the cesium concentrations. As before, the green diamonds mark locations

of those stations, P10-114 (42.17° N, 143.8° E), P10-112 (41.75° N, 144.13° E), P10-110 (41.25° N, 144.51° E), P10-108 (40.76° N, 144.88° E), P10-106 (40.08° N, 145.37° E) and P10-104 (39.42° N, 145.85° E), where the cesium concentrations in surface seawater samples have been measured by Kumamoto et al. (2014) to be at the background level.

The stations, P10-102 (38.75° N, 146.32° E), P10-100 (38.08° N, 146.77° E), P10-98 (37.42° N, 147.2° E), P10-96 (36.74° N, 147.63° E) and P10-94 (36.08° N, 148.05° E), where the concentrations have been found to be larger (but not exceeding $25.19 \pm 1.24$ mBq/kg for $^{137}$Cs), are indicated by the magenta diamonds. It's worth to stress a good qualitative correspondence with our simulation results 10 months after the accident in the sense that stations with measured background level are in the area of Oyashio "blue" waters with low risk to be contaminated, whereas stations with comparatively high levels of radiocesium concentrations are in the area of the Fukushima-derived "yellow" waters with increased risk of contamination.

As to the TsE, it was sampled later, in July 5, 2012, in the cruise of the R/V "Professor Gagarinskiy" (Budyansky et al., 2015) when it was a comparatively large mesoscale eddy around 150 km in diameter with the elliptic point at 41.3° N, 147.3° E consisting of intermittent strips of "blue" and "yellow" waters (Fig. 4d) which have been wrapped around during its growth from February to July 2012. Station 84 in that cruise was located near the elliptic point of that eddy (called as G by Budyansky et al. (2015)). The concentrations of $^{137}$Cs at the surface and at 100 m depth have been measured to be $11 \pm 0.6$ mBq/kg and $18 \pm 1.3$ mBq/kg, respectively, an order of magnitude larger than the background level. As to the $^{134}$Cs concentration, it was measured to be smaller, $6.1 \pm 0.4$ mBq/kg and $10.4 \pm 0.7$ mBq/kg due to a shorter half-lifetime of that isotope. In fact, it was one of the highest cesium concentrations measured inside all the eddy features sampled in the cruise 15 months after the accident.

The maximal concentration of radionuclides was observed, as expected, not at the surface but within subsurface and intermediate water layers (100–500 m) in the potential density range of 26.5–26.7 due to a convergence and subduction of surface water inside anticyclonic eddies. The corresponding tracking map in Fig. 10c by Budyansky et al. (2015) confirms its genetic link with the TE, and, therefore, a probability to detect increased cesium concentrations was expected to be comparatively large. We were able to track all the modification of the TsE up and its death on April 16, 2013 in the area around 40° N, 147.5° E.

## 4 Conclusions

We elaborated a specific Lagrangian methodology for simulating, tracking and documenting origin and history of water masses in ocean mesoscale features. Integrating advection equations for passive particles in the AVISO velocity

field backward in time, we have computed Lagrangian maps demonstrating clearly by which waters the mesoscale eddies in the Kuroshio – Oyashio confluence zone were composed of. It allowed to simulate by which ways they gained and lost water with a risk to be contaminated by Fukushima-derived radionuclides. We have studied three genetically-linked persistent mesoscale anticyclonic eddies in the area, TE, HE and TsE, which have been sampled in the R/V cruises in 2011 and 2012 to contain higher concentrations of radiocesium isotopes. The simulated Lagrangian maps allowed to document and analyze how they interact to pass radioactive water to each other. The simulated results have been shown to be in a good qualitative correspondence as compared with *in situ* measurements.

We hope that the proposed methodology could be applied to simulate propagation of pollutants after future possible accidents and identify and track contaminated persistent features in the ocean. The Lagrangian methodology seems to be useful, as well, to plan courses of the R/V cruises. It allows not only to track mesoscale eddies in the studied area but to identify the origin of water masses and to estimate *a priori* concentrations of radionuclides, pollutants or other Lagrangian tracers inside the eddies planned to be sampled.

*Acknowledgements.* The methodological part of the work was supported by the Russian Foundation for Basic Research (project no. 16–05–00213) and the simulations were supported by the Russian Science Foundation (project no. 16–17–10025). The altimeter products were distributed by AVISO with support from CNES.

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
