# Peer review of "Lagrangian simulation and tracking of the mesoscale eddies contaminated by Fukushima-derived radionuclides"

_Ocean Science, 2016_

## Referee Comment (RC1) · Anonymous Referee #1 · 25 Feb 2017

- Summary

This paper presents Lagrangian maps that visualize the origin, history and fate of the water masses in the mesoscale eddies off the coast of Japan. From the results, the authors argued: 1) the potential risk of contaminated water derived from the Fukushima nuclear power plant, 2) transition of water properties included in the mesoscale eddies and 3) qualitative correspondence between the Lagrangian maps with the observed Cs-137 data (Buesseler et al., 2012, Kaeriyama et al., 2013). The methodology is very interesting and the authors arguments 1) and 2) are understandable. However, the argument 3) is not supported by the results presented by this work. Paragraphs describing comparison with the observed data are just repeating the arguments of

their previous works (Prants et al., 2014 and Budyansky et al., 2015). The argument 3) is very important if the authors discuss about the potential risk of contaminated water (argument 1). Consequently, I recommend to reinforce the argument 3) with their results presented in this work. The concrete problems regarding the argument 3) are described below.

- Problems regarding the argument 3)

p. 9 L. 32-p. 10 L. 7: This paragraph is totally describing results by Prants et al. (2014) and not by this paper. The observed data by Kaeriyama et al. (2013) showed especially high Cs-137 concentrations in the green segment of Fig. 2d. The result of this study showed that the green segment is corresponding to the blue color water mass (Fig. 2d), which means Cs-137 concentrations are low. The authors need to address this inconsistence and discuss possible reasons explaining it.

p. 10 L. 26-29: Kaeriyama's transect is not reaching to "yellow water"; I do not agree with the authors conclusion is supported by the observation by Kaeriyama et al. (2013).

p. 11 L. 4-13: This paragraph is describing results by Budyansky et al. (2015). There are no discussion for correspondence between the Lagrangian map with the observed data.

- Other isuues

p. 9 L. 11: The meridional transect by Buesseler et al. (2012) is 144E, not 145E.

It is strongly recommended to add figures comparing the Lagrangian maps with the observed data. The comparisons are described in text, but they are hard to understand as readers need to look around the papers Buesseler et al. (2012) and Kaeriyama et al. (2013). Their data are publicly available and number of the data are not so many, it is easy to make figures to compare the Laglangian maps with the observed data.

---

## Referee Comment (RC2) · Anonymous Referee #2 · 27 Feb 2017

General comments

The authors used altimetry-derived surface geostrophic currents, to performed a series of Lagrangian experiments in the north-western Pacific Ocean, East of Japan, in the months following the Fukushima Nuclear Power Plant (FNPP) accident. They produced a series of successive maps in which each Lagrangian particle is back-tracked for two years and labelled according to its region of origin. The maps are used to infer the origin of the material associated with various mesososcale eddies identified in the region of study, and to investigate how the presence of such structures affected the dispersion of the Fukushima radioactive material after the accident. The results are compared with already published in-situ observations from three different cruises

showing good agreement.

Overall I found the paper to be clear and the figures of good quality. Although being too descriptive, the results presented provide good support for the interpretation of the in-situ observations and could be eventually worth publication. However, I do not recommend the paper to be published in its current state.

Major comments

As it is, a large portion of the paper is dedicated to the description of the results from previous ocean campaigns which have been already described in previous publications (Buessler et al. 2012; Prants et al. 2014; Kaeriama et al. 2013). The originality of the manuscript needs to be improved. For instance, the authors should provide more details on both a) the computation of Lagrangian diagnostics; b) the eddy identification and tracking. Regarding the Lagrangian diagnostics: little is said other than that they are derived from AVISO velocity fields. Are the trajectories derived using time varying fields? Are the velocity fields interpolated in both sapce and time? If so, how? At which spatial resolution are the Lagrangian particles used to generate the maps in Figures 2 to 4 deployed? Which type of AVISO product was used (dt or nrt; two or all sat merged)? On lines 95 to 107 the authors list a series of what they call "Lagrangian indicators", however, they are never shown in the following section. My suggestion is to include only the ones that are discussed in the rest of the manuscript. Among those the authors briefly mention the Finite-time Lyapunov exponent (FTLE). Again, if such diagnostic is used for any of the analyses described in the manuscript (i.e. for definign the elliptic and hyperbolic points introduced in section 2?), then more details on the background of the FTLE and the way they are defined and computed should be provided. More details should also be provided on how the elliptic and hyperbolic points are identified. In the manuscript (lines 163-165) they are simply defined as points of zero velocity. How is the circular and diverging/converging motion around those points identified? Regarding eddy detection and tracking, even less is provided. If they have been identified from previous studies, it should be explicitly said. Otherwise, more

details on the methodology should be given.

A second major issue is with the interpretation of the results from Figures 2 to 4. If I interpret them correctly, each point in the plot has been advected backward in time for two years and then color coded according to its region of origin. The regions are defined in figure 1a. White points are the ones that do not originate from any of those regions. I do agree with the authors, that such backtracking is quite powerful for understanding the origin of waters trapped within the 3 mesoscale eddies investigated. However, a 2-year backtracking means also that a portion of the "yellow" waters in the figures could have originated from the FNPP area from dates earlier than the accident, and thus not being necessarily contaminated with radioactive material. Wouldn't a Lagrangian experiment forward in time from the region of the FNPP after the time of the accident provide more suited trajectories/diagnostics to infer the fate of the radioactive material released? Specific of Figure 2a, at 144 E between 36 and 38 N, there is a very sharp zonal boundary between the different water masses. To me it appears similar to an artefact (either from data processing or visualization). Can the authors comments/correct that?

A final issue regards the methodological approach: in several occasion throughout the manuscript the authors refer to their Lagrangian approach as "special" (e.g. line 433). Although I agree with the authors when they claim that this type of diagnostics provide a more condensed, easier to read/interpret information, compared to spaghetti plots of particle trajectories, the approach they propose is not entirely novel, as several studies in recent years (for instance d'Ovidio et al. 2015, Biogeoscience to cite one of the most recent ones) have been based on the analysis of similar Lagrangian diagnostics. I suggest to rephrase some of the sentences describing the approach to make this clearer.

Furthermore, it is not clear to me how the presented approach would improve the limitations and uncertainties of deriving Lagrangian trajectories in a chaotic environment (lines 71 to 74). In the paragraphs from lines 75 to 85, the authors seem to hint to that

the adopted Lagrangian approach is more robust because it does not require "a precise solution of the Navier-Stokes equation". However, any Lagrangian diagnostic is based on trajectories which require a velocity field to be derived. Later in the manuscript, the author further remark the robustness of their approach stating that it is based on a "statistically significant number of particles". However, other than a large number of particles, shouldn't a statistical Lagrangian approach include also a random-walk term of some sort to simulate sub-grid diffusive-like processes? In my opinion, increasing the number of advected particles, while maintaining the same velocity field and the same equations as a coarser experiment, will provide more detailed results, but not significantly different than a coarser experiment (see for instance Hernández-Carrasco et al., 2011, Ocean Modelling). These two aspects should be clarified by the authors.

Minor comments

- The term "Lagrangian particles" should replace "tracers" in several instances in the manuscript.

- Line 66: if possible I would add some references to the Lagrangian studies which focused on the Horizon accident in the Gulf of Mexico. I am sure that not all of them were only based on the analysis of "spaghetti-like" plots.

- Fig 1a: I would plot SST rather than velocity magnitude (since temperature variation and fronts are repeatedly used in the introduction), and I would remove the indication of elliptic and hyperbolic points since in a climatological velocity field they do not represent mesoscale structures.

- Line 79: Diffusion will always occur. It is the mixing induced by advection that is reduced across transport barriers.

- Line 123: replace "twofold" with "threefold" since the same paragraph contain a "Firstly...", a "Secondly..." and a "Finally..."

- Line 147-149: AVISO provide the geostrophic component of the real near-surface

velocities.

- Line 165-167: The sentence should be moved after line 169, since it refers to elliptic points only only.

- Line 176: Several studies in the last few years have shown indeed that LCS and hyperbolic points can be identified and tracked for several days from in-situ observations: Haza et al. 2010, Ocean Dynamics; Nencioli et al 2011, JGR-Oceans; Olascoaga et al. 2013, GRL. They should be cited here.

- Line 381: were the ARGO floats regular float, or were characterized by specific configurations?

―――――――――――――――――――――――

---

## Author Comment (AC2) · 23 Mar 2017

**The author's respond to the Reviewer #1**

We are very grateful to the Reviewer for a very careful reading of the manuscript and a number of useful comments and critics we tried to take into account in the revised version. Our point-to-point responses can be found below.

**The main changes made**

We have modified our simulation code and recalculated all the results with this new code. The code, used in the previous version of our manuscript (ms) for plotting figures, underestimated the amount of yellow tracers from the area around the Fukushima Nuclear Power Plant (FNPP) advected outside that area and had a high risk to be contaminated. As before, we distribute a large number of particles in a large area in the northwestern Pacific on a fixed date and advected them backward in time. In the previous code, the particles, which crossed the yellow rectangular around the FNPP (Fig.1a) in the past for the period from the day of accident, March 11, 2011 to May 18, 2011, have been marked by the yellow color on the corresponding Lagrangian map. The particles, which were present in that area and leaved it after May 18, have not been colored in yellow. However, those particles also have a risk to be highly contaminated and should be specified as yellow ones. The present code specifies all those particles as yellow ones. As the result, some "white" waters, which have not been specified previously, now have been specified to came from the yellow area around the FNPP with a high risk to be contaminated.

We've cardinally rewritten Secs. 3.1, 3.2 and 3.3 to compare more clearly our simulation results with the measurements by Buesseler2012, Kaeriyama2013, Kuramoto2014 and Budyansky2015. With this aim we imposed on Figs.2c, 2d, 4a and 4d locations of stations with measured values of the cesium concentration levels in collected surface seawater samples in 2011 and 2012. For convenience, we place an updated version of the main Sec.3 'Results' to the end of the respond along with figures being changed.

When working on a revised version of our ms, we have found the paper by Kumamoto, Y. et al. Southward spreading of the Fukushima-derived radiocesium across the Kuroshio Extension in the North Pacific. Sci. Rep. 4, 4276; DOI:10.1038/srep04276 (2014). Seawater samples for radiocesium measurements in the frontal area have been collected during the R/V 'Mirai' cruise in the very beginning of February 2012. We used this new possibility to compare our simulations with this new data. We imposed on the simulated Lagrangian map in Fig.4a locations of stations to the north of the Kuroshio Extention (>36N) with measured levels of the cesium concentrations and found a good qualitative correspondence of those measurements with our simulation results 10 months after the accident in the sense that stations with measured background level are in the area of Oyashio ``blue" waters with low risk to be contaminated, whereas stations with comparatively high level of radiocesium concentrations are in the area of Fukushima-derived ``yellow" waters with increased risk of contamination.

**I. Reviewer #1: Recommendation -**

This paper presents Lagrangian maps that visualize the origin, history and fate of the water masses in the mesoscale eddies off the coast of Japan. From the results, the authors argued: 1) the potential risk of contaminated water derived from the Fukushima nuclear power plant, 2) transition of water properties included in the mesoscale eddies and 3) qualitative correspondence between the Lagrangian maps with the observed Cs-137 data (Buesseler et al., 2012, Kaeriyama et al., 2013).

The methodology is very interesting and the authors arguments 1) and 2) are understandable. However, the argument 3) is not supported by the results presented by this work. Consequently, I recommend to reinforce the argument 3) with their results presented in this work. The concrete problems regarding the argument 3) are described below.

**Responses to the First Reviewer's report (Problems regarding the argument 3) 1. Cited from the referee's report**

p. 9 L. 32-p. 10 L. 7: This paragraph is totally describing results by Prants et al. (2014) and not by this paper. The observed data by Kaeriyama et al. (2013) showed especially high Cs-137 concentrations in the green segment of Fig. 2d. The result of this study showed that the green segment is corresponding to the blue color water mass (Fig.2d), which means Cs-137 concentrations are low. The authors need to address this inconsistence and discuss possible reasons explaining it.

**Our response**

We removed that paragraph from the revised test. We recalculated all the results and imposed on the updated version of Fig. 2d locations of stations in the end of July 2011 with measured values of the cesium concentration levels in collected surface seawater samples by Kaeriyama et al. (2013). We removed colored segments in updated version of Fig.2. The green segment in former Fig. 2d corresponds to stations 43 (41N), 44 (40.5N), 45 (40N) and 46 (39.5N) marked by the magenta diamonds in the updated version of Fig. 2d.

'Station 43 was located inside the anticyclone HE filled mainly by ``yellow'' waters, and we estimate the risk to found Fukushima-derived radionuclides there to be large. Station 44 was located at the southern periphery of the anticyclone HE at the boundary between ``white'' and ``blue'' waters but in close proximity to a ``yellow'' streamer.' (cited from the updated version of Sec.3.2).

As to stations 45 and 46, the following text has been added to the updated version of Sec.3.1:

'However, there is an inconsistence of simulation with samplings at stations 45 and 46 where there are practically now yellow tracers but only blue and white ones. The reasons of this inconsistence might be different. In this paper we track only those tracers which were originated from the blue, red and black segments and the yellow rectangular around the FNPP shown in Fig.1a. So we did not specify the origin of white waters. They could reach their places on the maps from anywhere besides those segments and the area around the FNPP. They could in principle contain Fukushima-derived radionuclides deposited at the sea surface from the atmosphere after the accident and then being advected by eddies and currents in the area. Moreover, they could be those tracers which have been located inside AVISO grid cells near the coast around the FNPP just after the accident and then have been advected outside. We removed from consideration all the tracers entered into any AVISO grid cell with two or more corners touching the land because of inaccuracy of the altimetry-based velocity field there and in order to avoid artifacts. Thus, the white streamers inside the core and at the peripherv of the blue cyclone with the center at  $N{39.7}$ ,  $E{144.2}$  at the places or nearby stations 45 and 46 with high measured concentrations of cesium by \citep{Kaeriyama13}, could, in principle, contain contaminated water. However, it has not been proved in our simulation by the mentioned reasons' (cited from the updated version of Sec.3.1).

**2. Cited from the referee's report**

p. 10 L. 26-29: Kaeriyama's transect is not reaching to "yellow water"; I do not agree with the authors conclusion is supported by the observation by Kaeriyama et al. (2013).

**Our response**

As it is clearly seen in the updated version of Fig. 2d, the Kaeriyama's transect reaches to 'yellow' water. Its northern station 43 was located inside the anticyclone HE filled mainly by 'yellow'' waters, and we estimate the risk to found Fukushima-derived radionuclides there to be large. It is supported by the measurements of concentration of Cs 137 by Kaeriyama et al. (2013) to be around 71 mBq/kg at that station.

**3. Cited from the referee's report**

p. 11 L. 4-13: This paragraph is describing results by Budyansky et al. (2015). There are no discussion for correspondence between the Lagrangian map with the observed data.

**Our response**

Our ms is intended for a wide audience, not only to specialists in radioactivity measurements. The corresponding paragraph really describes mainly some results by Budyansky et al. (2015). We put it to the context because the measurements by Budyansky et al. (2015) and some simulation results in that paper support our conclusions about the role of the Tsugaru eddy (TsE) in transport and mixing of Fukushima-derived radionuclides. Moreover, we imposed on updated version of Fig. 4d location of station 84 with measured increased values of the cesium concentration levels in July2012, 15 months after the accident.

**Other isuues**

**4. Cited from the referee's report**

p. 9 L. 11: The meridional transect by Buesseler et al. (2012) is 144E, not 145E.

**Our response**

Thank you, it's our mistake. The meridional transect by Buesseler et al. (2012) is shown in Fig.2c and in its caption to be along 144E, not 145E. We corrected this mistake in the main text.

**5.** Cited from the referee's report**

It is strongly recommended to add figures comparing the Lagrangian maps with the observed data. The comparisons are described in text, but they are hard to understand as readers need to look around the papers Buesseler et al. (2012) and Kaeriyama et al. (2013). Their data are publicly available and number of the data are not so many, it is easy to make figures to compare the Laglangian maps with the observed data.

**Our response**

We have done that and imposed on Figs.2c, 2d and 4d locations of stations with measured values of the cesium concentration levels in collected surface seawater samples in 2011 and 2012. We thank again the Reviewer for advising that. It helped us not only to improve the ms but forced us to recalculate all the results.

**The updated version of the main Sec.3 'Results' along with figures being changed**

[revised manuscript text omitted]

---

## Author Comment (AC1)

**The author's respond to the Reviewer #2**

We are very grateful to the Reviewer for a very careful reading of the manuscript and a number of useful comments and critics we tried to take into account in the revised version. Our point-to-point responses can be found.

**The main changes made**

We have modified our simulation code and recalculated all the results with this new code. The code, used in the previous version of our manuscript (ms) for plotting figures, underestimated the amount of yellow tracers from the area around the Fukushima Nuclear Power Plant (FNPP) advected outside that area and had a high risk to be contaminated. As before, we distribute a large number of particles in a large area in the northwestern Pacific on a fixed date and advected them backward in time. In the previous code, the particles, which crossed the yellow rectangular around the FNPP (Fig.1a) in the past for the period from the day of accident, March 11, 2011 to May 18, 2011, have been marked by the yellow color on the corresponding Lagrangian map. The particles, which were present in that area and leaved it after May 18, have not been colored in yellow. However, those particles also have a risk to be highly contaminated and should be specified as yellow ones. The present code specifies all those particles as yellow ones. As the result, some "white" waters, which have not been specified previously, now have been specified to came from the yellow area around the FNPP with a high risk to be contaminated.

We've cardinally rewritten Secs. 3.1, 3.2 and 3.3 to compare more clearly our simulation results with the measurements by Buesseler2012, Kaeriyama2013, Kuramoto2014 and Budyansky2015. With this aim we imposed on Figs.2c, 2d, 4a and 4d locations of stations with measured values of the cesium concentration levels in collected surface seawater samples in 2011 and 2012.

When working on a revised version of our ms, we have found the paper by Kumamoto, Y. et al. Southward spreading of the Fukushima-derived radiocesium across the Kuroshio Extension in the North Pacific. Sci. Rep. 4, 4276; DOI:10.1038/srep04276 (2014). Seawater samples for radiocesium measurements in the frontal area have been collected during the R/V 'Mirai' cruise in the very beginning of February 2012. We used this new possibility to compare our simulations with this new data. We imposed on the simulated Lagrangian map in Fig.4a locations of stations to the north of the Kuroshio Extention (>36N) with measured levels of the cesium concentrations and found a good qualitative correspondence of those measurements with our simulation results 10 months after the accident in the sense that stations with measured background level are in the area of Oyashio ``blue" waters with low risk to be contaminated, whereas stations with comparatively high level of radiocesium concentrations are in the area of Fukushima-derived ``yellow" waters with increased risk of contamination.

**Reviewer #2: Recommendation -**

Overall I found the paper to be clear and the figures of good quality. Although being too descriptive, the results presented provide good support for the interpretation of the in-situ observations and could be eventually worth publication. However, I do not recommend the paper to be published in its current state.

**Responses to the Second Reviewer's report (Major comments)**

**1. Cited from the referee's report**

As it is, a large portion of the paper is dedicated to the description of the results from previous ocean campaigns which have been already described in previous publications (Buessler et al. 2012; Prants et al. 2014; Kaeriama et al. 2013). The originality of the manuscript needs to be improved. For instance, the authors should provide more details on both a) the computation of Lagrangian diagnostics; b) the eddy identification and tracking.

**Our response**

We removed from the revised test in Sec.3.1. all the paragraph describing our previous results (Prants et al. 2014). We included in the text a short description of in situ measurements of concentration of Fukushima-derived radiocesium in 2011 and 2012 and impose locations of corresponding stations in the cruises by Buessler, Kaeriama and Budyansky for ease of comparison of observations with our simulation. The computation of Lagrangian diagnostics is described in the second part of Sec.2 Data and methodology on the whole page 3. We have added there the following text describing the eddy identification and tracking as the reviewer asked:

'The altimetry-based Lagrangian maps allow accurately identify and track mesoscale eddies and document their transformation due to interactions with currents and other eddies. Inspecting daily-computed Lagrangian maps for a long period of time (for two years in this paper) and computing stagnation elliptic points daily, one can track the fate of any eddy if it is sufficiently large and long lived (more than a week). The Lagrangian diagnostics is more appropriate with that aim than the commonly used techniques because the Lagrangian maps are imprints of history of water masses involved in the vortex motion whereas vorticity, the Okubo-Weiss parameter and similar indicators are ``instantaneous'' snapshots (see Ref.~\citep{Prants2016} for comparison).'

**2. Cited from the referee's report**

Regarding the Lagrangian diagnostics: little is said other than that they are derived from AVISO velocity fields. Are the trajectories derived using time varying fields? Are the velocity fields interpolated in both sapce and time? If so, how?

**Our response**

We've edited the text in the beginning of Sec.2 Data and methodology as follows:

'All the simulation results are based on integrating equations of motion for a large number of synthetic particles (tracers) advected by the AVISO velocity field  $\frac{d\lambda}{dt} = u(\lambda, \varphi, t), \frac{d\varphi}{dt} = v(\lambda, \varphi, t)$ , where u and v are angular zonal and meridional velocities,  $\varphi$  and  $\lambda$  are latitude and longitude, respectively. The altimetry-based velocities were obtained from the AVISO database ( $url{aviso.altimetry.fr}$ ) archived daily on a  $1/4^{\circ} \times 1/4^{\circ}$  grid. The velocity field has been interpolated using a bicubical spatial interpolation and third order Lagrangian polynomials in time. In integrating Eqs.~\ref{adveq} we used a fourth-order Runge-Kutta scheme with an integration step of 0.001 day.'

**3. Cited from the referee's report**

At which spatial resolution are the Lagrangian particles used to generate the maps in Figures 2 to 4 deployed?

**Our response**

To generate the maps in Figures 2 to 4 we used 700x700 Lagrangian particles. It is mentioned in the revised text.

**4. Cited from the referee's report**

Which type of AVISO product was used (dt or nrt; two or all sat merged)?

**Our response**

The AVISO product 'all sat merged delayed time' was used.

**5. Cited from the referee's report**

On lines 95 to 107 the authors list a series of what they call "Lagrangian indicators", however, they are never shown in the following section. My suggestion is to include only the ones that are discussed in the rest of the manuscript.

**Our response**

The manuscript is intended for a wide audience, not only to experts in Lagrangian diagnostics. We prefer to save a single sentence listing different Lagrangian indicators because they could be used to provide additional Lagrangian information on transport and mixing of passive particles and tracers.

**6. Cited from the referee's report**

Among those the authors briefly mention the Finite-time Lyapunov exponent (FTLE). Again, if such diagnostic is used for any of the analyses described in the manuscript (i.e. for definign the elliptic and hyperbolic points introduced in section 2?), then more details on the background of the FTLE and the way they are defined and computed should be provided.

**Our response**

We did not apply the FTLE technique in this manuscript and removed mention of the FTLE in the revised text.

**7. Cited from the referee's report**

More details should also be provided on how the elliptic and hyperbolic points are identified. In the manuscript (lines 163-165) they are simply defined as points of zero velocity. How is the circular and diverging/converging motion around those points identified?

**Our response**

We have added the following text to clarify that:

**'The elliptic points are called stable and the hyperbolic ones are unstable. Their local stability properties are characterized by a standard method by eigenvalues of the Jacobian matrix of the velocity field.'**

The circular and diverging/converging motion around those points are also defined by eigenvalues of the Jacobian matrix.

**8. Cited from the referee's report**

Regarding eddy detection and tracking, even less is provided. If they have been identified from previous studies, it should be explicitly said. Otherwise, more details on the methodology should be given.

**Our response**

We have added there a short text describing the eddy identification and tracking as the reviewer asked.

'The altimetry-based Lagrangian maps allow accurately identify and track mesoscale eddies and document their transformation due to interactions with currents and other eddies. Inspecting daily-computed Lagrangian maps for a long period of time (for two years in this paper) and computing stagnation elliptic points daily, one can track the fate of any eddy if it is sufficiently large and long lived (more than a week). The Lagrangian diagnostics is more appropriate with that aim than the commonly used techniques because the Lagrangian maps are imprints of history of water masses involved in the vortex motion whereas vorticity, the Okubo-Weiss parameter and similar indicators are ``instantaneous'' snapshots (see Ref.~\citep{Prants2016} for comparison).'

**9. Cited from the referee's report**

A second major issue is with the interpretation of the results from Figures 2 to 4. If I interpret them correctly, each point in the plot has been advected backward in time for two years and then color coded according to its region of origin. The regions are defined in figure 1a. White points are the ones that do not originate from any of those regions. I do agree with the authors, that such backtracking is quite powerful for understanding the origin of waters trapped within the 3 mesoscale eddies investigated.

However, a 2-year backtracking means also that a portion of the "yellow" waters in the figures could have originated from the FNPP area from dates earlier than the accident, and thus not being necessarily contaminated with radioactive material. Wouldn't a Lagrangian experiment forward in time from the region of the FNPP after the time of the accident provide more suited trajectories/diagnostics to infer the fate of the radioactive material released?

**Our response**

Cited from the manuscript (p.3, lines 187-190): 'In what follows we specify on the maps "yellow waters" as those which have a large risk to be contaminated because they came from the area just around the FNPP enclosed by the yellow lines in Fig. 1a for the period from the day of the accident, March 11, 2011, to May 18, 2011 when direct releases of radioactive isotopes to the ocean and atmosphere stopped'. So, we did not track waters from dates earlier than the accident.

**10. Cited from the referee's report**

Specific of Figure 2a, at 144 E between 36 and 38 N, there is a very sharp zonal boundary between the different water masses. To me it appears similar to an artefact (either from data processing or visualization). Can the authors comments/correct that?

**Our response**

We have modified our simulation code and recalculated all the results with this new code. In updated Fig. 2a the straight zonal boundary along 36.5 N and meridional boundary along 144 E, separating water masses of different origin, are just fragments of the yellow straight lines in Fig.1a restricting a potentially radioactive area around the FNPP. The map in Fig. 2a corresponds to March 26, 2011, only 15 days after the accident. The yellow color marks the tracers that have been inside that area or leaved it for these 15 days. The waters of the other colors near those straight zonal and meridional boundaries moved inside the area. So, these boundaries separate the 'yellow' tracers which were present within the area from those ones which have not yet managed to penetrate inside the area for the 15 days.

We briefly clarified that in the revised version to avoid misunderstanding.

**11. Cited from the referee's report**

A final issue regards the methodological approach: in several occasion throughout the manuscript the authors refer to their Lagrangian approach as "special" (e.g. line 433). Although I agree with the authors when they claim that this type of diagnostics provide a more condensed, easier to read/interpret information, compared to spaghetti plots of particle trajectories, the approach they propose is not entirely novel, as several studies in recent years (for instance d'Ovidio et al. 2015, Biogeoscience to cite one of the most recent ones) have been based on the analysis of similar Lagrangian diagnostics. I suggest to rephrase some of the sentences describing the approach to make this clearer.

**Our response**

By the word "special" we mean not 'novel' or 'new' but just a thing 'of a particular or certain sort' (cited from the Oxford Dictionary). For example, a special train is an extra train for special purposes.

**12. Cited from the referee's report**

Furthermore, it is not clear to me how the presented approach would improve the limitations and uncertainties of deriving Lagrangian trajectories in a chaotic environment (lines 71 to 74). In the paragraphs from lines 75 to 85, the authors seem to hint to that the adopted Lagrangian approach is more robust because it does not require "a precise solution of the Navier-Stokes equation".

However, any Lagrangian diagnostic is based on trajectories which require a velocity field to be derived. Later in the manuscript, the author further remark the robustness of their approach stating that it is based on a "statistically significant number of particles". However, other than a large number of particles, shouldn't a statistical Lagrangian approach include also a random-walk term of some sort to simulate sub-grid diffusive-like processes? In my opinion, increasing the number of advected particles, while maintaining the same velocity field and the same equations as a coarser experiment, will provide more detailed results, but not significantly different than a coarser experiment (see for instance Hernandez-Carrasco et al., 2011, Ocean Modelling). These two aspects should be clarified by the authors.

**Our response**

In this manuscript we don't claim that 'the Lagrangian approach is more robust than a precise solution of the Navier-Stokes equation' (cited from the referee's report). We only claim along with many other authors cited that the Lagrangian approach allows to find more or less robust material structures in chaotic flows without a precise solution of the Navier-Stokes equations (see the first paragraph on p.2, the right column).

We agree, in principle, with the referee that 'increasing the number of advected particles, while maintaining the same velocity field and the same equations as a coarser experiment, will provide more detailed results, but not significantly different than a coarser experiment' (cited from the referee's report). However, obtaining 'more detailed results' is crucial for a specific problem we consider in this ms, tracking of Fukushima-contaminated waters and comparing the simulation results with observation ones.

As to introducing 'a random-walk term of some sort to simulate sub-grid diffusive-like processes' (cited from the referee's report). For chaotic systems, the introduction of a random-walk term in the equations of motion seems to make little sense, because Lyapunov instability of trajectories quickly amplifies noise of any computation method (which is always present), and of a finite accuracy of the representation of real numbers.

**Minor comments**

**13. Cited from the referee's report**

The term "Lagrangian particles" should replace "tracers" in several instances in the manuscript.

**Our response**

We remained the term "tracers" unchanged when speaking about 'colored' particles originated from specified places and replaced it by "Lagrangian particles" in the other cases.

**14. Cited from the referee's report**

Line 66: if possible I would add some references to the Lagrangian studies which focused on the Horizon accident in the Gulf of Mexico. I am sure that not all of them were only based on the analysis of "spaghetti-like" plots.

**Our response**

This and the next paragraphs have been slightly edited for more clarity and the following references have been added:

Mezic2010: Mézic IS, Loire VAF, Hogan P (2010) A new mixing diagnostic and the Gulf of Mexico oil spill. Science 330:489.

Olascoaga2012: M. J. Olascoaga and G. Haller}, Forecasting sudden changes in environmental pollution patterns, Proc. National Academy of Sciences, 2012, v.109, p.4738.

Huntley2011: Huntley HS, Lipphardt B, Kirwan A (2011) Surface drift predictions of the deepwater horizon spill: The Lagrangian perspective. Geophys Monogr Ser 195:179–195.

**The text now reads as follows:**

'The standard approach in simulating transport phenomena like propagation of oil after the explosion at the Blue Horizon mobile drilling rig in the Gulf of Mexico in April 2010 and propagation of radioactive isotopes after the accident at the FNPP is to run global or regional numerical models of circulation to simulate propagation of pollutants and try to forecast their

trajectories. The outcomes provide ``spaghetti-like'' plots of individual trajectories which are hard to interpret. Moreover, majority of trajectories in a chaotic environment are very sensitive to small and inevitable variations in initial conditions. Those trajectories are practically unpredictable even over a comparatively short time.

The specific Lagrangian approach, based on dynamical systems theory, has been developed in the last decades with the aim to find more or less robust material structures in chaotic flows governing mixing and transport of Lagrangian particles and creating transport barriers of a contaminant across them preventing propagation \citep[for reviews see][] {Samelson,MS06,KP06,Haller ARFM2015}. Identification of such structures in the ocean would help to predict for a short and medium time where a contaminant will move even without a precise solution of the Navier\mdash Stokes equations. This approach has been successfully used simulating propagation in Gulf in of oil the Mexico of \citep{Mezic2010,Huntley2011,Olascoaga2012} and propagation of Fukushima-derived radionuclides in the Pacific ocean \citep{DAN11,DSR201,Prants2014}.'

**15. Cited from the referee's report**

Fig 1a: I would plot SST rather than velocity magnitude (since temperature variation and fronts are repeatedly used in the introduction), and I would remove the indication of elliptic and hyperbolic points since in a climatological velocity field they do not represent mesoscale structures. **Our response**

Figure 1a with the averaged AVISO velocity field from 1993 to 2016 gives an image of persistent mesoscale features in the study area governing the large-scale transport and mixing. SST does not provide that. We removed the indication of elliptic and hyperbolic points from that figure.

**16. Cited from the referee's report**

Line 79: Diffusion will always occur. It is the mixing induced by advection that is reduced across transport barriers.

**Our response**

We mean there a diffusive-like propagation, not a molecular diffusion. However, in order to avoid a misunderstanding we removed the term 'diffusive-like' from the text and edited the sentence as:

'The specific Lagrangian approach, based on dynamical systems theory, has been developed in the last decades with the aim to find more or less robust material structures in chaotic flows governing mixing and transport of Lagrangian particles and creating transport barriers preventing propagation of a contaminant across them.'

**17. Cited from the referee's report**

Line 123: replace "twofold" with "threefold" since the same paragraph contain a "Firstly...", a "Secondly..." and a "Finally..."

**Our response**

Done.

**18. Cited from the referee's report**

Line 147-149: AVISO provide the geostrophic component of the real near-surface velocities. **Our response**

Done.

**19. Cited from the referee's report**

Line 165-167: The sentence should be moved after line 169, since it refers to elliptic points only.

**Our response**

Done.

**20. Cited from the referee's report**

Line 176: Several studies in the last few years have shown indeed that LCS and hyperbolic points can be identified and tracked for several days from in-situ observations: Haza et al. 2010, Ocean Dynamics; Nencioli et al 2011, JGR-Oceans; Olascoaga et al. 2013, GRL. They should be cited here.

**Our response**

The following sentence has been added just after Lone 176:

'The hyperbolic points and their attracting and repelling manifolds have been recently identified with the help of drifter's tracks in the Gulf of La Spezia in the northwestern Mediterranean Sea \citep{Haza2010}, in the Gulf of Mexico \citep{Olascoaga2013} and in the northwestern Pacific \citep{Prants2016}.'

Haza2010: Haza, A.C., Özgökmen, T.M., Griffa, A. et al. Transport properties in small-scale coastal flows: relative dispersion from VHF radar measurements in the Gulf of La Spezia. Ocean Dynamics (2010) 60: 861. doi:10.1007/s10236-010-0301-7

Olascoaga2013: M. J. Olascoaga et al. Drifter motion in the Gulf of Mexico constrained by altimetric Lagrangian coherent structures. GEOPHYSICAL RESEARCH LETTERS, V. 40, doi:10.1002/2013GL058624, 2013.

Prants2016: S.V. Prants, V.B. Lobanov, M.V. Budyansky, M.Yu. Uleysky. Lagrangian analysis of formation, structure, evolution and splitting of anticyclonic Kuril eddies. Deep Sea Research I. V.109 pp.61–75 (2016). DOI: 10.1016/j.dsr.2016.01.003

**21. Cited from the referee's report**

Line 381: were the ARGO floats regular float, or were characterized by specific configurations?

**Our response**

They are regular floats. We corrected the text

[revised manuscript text omitted]

---

## Author Response (AR2)

**The 2nd author's respond to the Reviewer #1**

**I. Cited from the referee's report**

The authors have revised the manuscript adequately. I recommend to publish this manuscript as is. Only one thing that I would like to recommend to revise is to indicate the concrete threshold value separating the green and magenta diamonds used in Figs. 2 and 4.

**Our response**
**We did that correcting the last sentence in Sec.3 on page 5 (line 5) to be**
**'For ease of comparison, we mark in Fig.~\ref{figTE}c by the green diamonds the locations of stations 13 and 14 with collected surface seawater samples by \citet{Buesseler12} in which the cesium concentrations have been measured to be at the background level ($\lesssim 3.6$~mBq/kg).'**

**The 2nd author's respond to the Reviewer #2**

We are very grateful to the referee for useful comments and suggestions.

**Responses to the Second Reviewer's report**
**1. Cited from the referee's report**
I recommend some minor (mostly stylistic) modifications before publication.

**Our response**
**We are very grateful to the referee for that job and corrected all the points indicated.**

**2. Cited from the referee's report**
Pag 3: I do not see why the reference Nencioli et al. 2011 should be left out of that list: it is the same type of analysis as Olascoaga et al., 2013, but from two years earlier!!! Please add it in the text.

**Our response**
**The paper by Nencioli et al. 2011 has been added to the list of references.**

**3. Cited from the referee's report**
Regarding the origin of the "yellow" waters (Pag 3 L95): My apologies to the authors for having misinterpreted the text. However, the sentence they cite in their response has been now removed from the text. I suggest to keep it: "...the yellow straight lines in Fig 1a, AND for the period from the day of the accident, March 11, 2011, to May 18, 2011, when direct releases of radioactive isotopes to the ocean and the atmosphere stopped." (by adding AND , the two distinct conditions required to mark an advected particle as yellow water are further remarked).

**Our response**
**We did that, thank you.**

**4. Cited from the referee's report**
Regarding the use of the adjective "special": To avoid any sort of confusion, I think that the adjective "specific" could be used, instead, in the following sentences: Pag 3 L81: "...in the area, in this paper, we develop a specific Lagrangian diagnostic oriented to..." Pag 10 L1: "We developed a specific Lagrangian methodology..."

**Our response**
**OK, we replaced 'special' by 'specific' in both the cases.**

**5. Cited from the referee's report**
Figure 2: It would make more sense if reference to figure 2 and 1 were swapped in the text:
L77 to 83: ...Fig 2a on March 26,2011 (…) lines in Fig.1a restricting… Moreover, the figure would be clearer if the boundary showed in fig 1 would be repeated again in Fig.2a.

 **Our response**
**We swapped the reference to figures 2 and 1 and showed the boundary in fig 2a by the dashed black line.**

[revised manuscript text omitted]

---

## Author Response (AR3)

Dear Dr. Matthew Hecht,

thank you for your suggestions.

We corrected the text appropriately and uploaded all the production files.

With best regards, authors